# The Changes in GABA, GAD and DAO Activities, and Microbial Safety of Soaking- and High Voltage Electric Field-Treated Adzuki Bean Sprouts

**Kai-Ying Chiu**

Department of Post-Modern Agriculture, MingDao University, Changhua 52345, Taiwan; cky@mdu.edu.tw;
Tel.: +886-0953-835-832

**Abstract:** The level of γ-aminobutyric acid (GABA) in nongerminated adzuki bean seeds is low, but it increases substantially during germination and sprouting. In this study, three seed treatments, including soaking (S), high voltage electric field (HVEF), and soaking plus HVEF (SHVEF), were used to examine their effects on sprout growth, sprout GABA content, sprout glutamate decarboxylase (GAD), and diamine oxidase (DAO) activities and microbial loads on 6-day-old adzuki bean sprouts. All the treatments enhanced sprout growth, increased sprout's GABA, and increased sprouts' GAD and DAO activities. The examined seed treatments also significantly reduced the microbial loads of the produced 6-day-old adzuki bean sprouts. The most effective treatment that improved the morphological and biochemical traits and reduced microbial loads on produced sprouts was the SHVEF treatment. SHVEF treatment also achieved a 5-log reduction in the microbial loads of total aerobic bacterial counts, total coliform counts, and total mold counts on the produced adzuki bean sprouts. Therefore, SHVEF is effective for increasing adzuki bean sprout production. It can also be used to improve nutritional quality and provide an intervention technique against microbial contamination on produced sprouts.

**Keywords:** adzuki bean; GABA; germination; HVEF; microbial load; sprout production

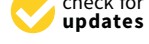



## 1. Introduction

Gamma-aminobutyric acid (GABA) is a naturally occurring amino acid present in plants, animals, and microorganisms. It is produced mainly from the cytosolic glutamate decarboxylase (GAD, EC 4.1.1.15) catalyzed glutamate decarboxylation [1,2]. The reaction catalyzed by GAD is the main route for GABA biosynthesis in herbaceous plants [1]. However, GABA can also be synthesized from the cell wall diamante oxidase (DAO, EC 1.4.3.22) catalyzed polyamine degradation [3], which is not detectable in quiescent seeds but appears during the early phase of germination [4]. GABA can also be produced by a nonenzymatic reaction from proline under oxidative stress [5]. In humans, GABA acts as a depressive neurotransmitter in the central nervous system, and it can also regulate blood pressure and relieve pain and anxiety [6]. In plants, GABA is associated with the metabolism of carbohydrates and amino acids [7,8], and the defense against various abiotic stresses [9]. GABA has recently received substantial attention and has become a very common dietary supplement due to its antianxiety, analgesic effects, and hypotensive activity [10]. Therefore, many efforts have been made to develop new technological processes for GABA enhancement in traditional foodstuffs or avoiding losses after processing treatments [11].

Sprout is the product obtained from the germination of plant seeds. It is frequently consumed as a ready-to-eat vegetable [12]. It is known that germination would bring about the accumulation of many bioactive compounds including polyphenols, vitamins, and GABA [13]. GABA in bean seeds is generally found at low level [14,15], nevertheless, the level rapidly increases during germination. Adzuki beans (*Vigna angularis*) have been widely cultivated for centuries in several Asian countries [16]. They are either directly

cooked as red bean soup or prepared as sweet paste. Moreover, the adzuki bean seeds are consumed as sprouts, owing to their crispy texture and attractive smell. Germination was reported to increase the level of GABA in adzuki bean sprouts, compared to the level of GABA obtained from nongerminated adzuki bean seeds [17].

Many methods have been developed for improving seed germination [18]. Among the practicable techniques, soaking is an effective and low-cost way to improve germination. During soaking, the seed's metabolism is activated in preparation for germination [19]. Soaking tended to affect the chemical composition, antinutritional factors, and nutritional quality of selected legume and cereal seeds [20]. Soaking-induced microstructure changes accounted for the losses of nutrients and antinutritional factors during soaking [21]. Soaking the adzuki bean seeds prior to germination was also reported to increase the GABA level in produced bean sprouts [22].

High-voltage electric field (HVEF) is a nonthermal technology frequently used in food industries [6]. Positive results in the germination improvement were also obtained with HVEF application. The use of HVEF treatment was reported to enhance the germination of chickpea (*Cicer arietinum*) and bitter gourd (*Momordica charantia*) seeds [23,24]. However, the electrical field treatment did not significantly increase the germination of treated carrot (*Daucus carota* L.) seeds [25]. Thus, the effectiveness of electrical field treatment, which is affected by several important factors such as electric field intensity and treatment duration, appears to be crop species dependent.

There has been an increasing trend in consumption of sprouts worldwide; therefore, sprout contamination has become a major concern [26]. However, soaking treatment showed little effect on reducing microbial population, even though it improved seed germination [27]. HVEF has also been widely used to maintain the quality of foods by reducing bacterial infection through HVEF-induced electroporation [28]. The objective of this study was to evaluate the ability of soaking (S), high-voltage electric field (HVEF) and soaking plus HVEF (SHVEF) in relation to the enhancement in seed germination and sprout growth for the tested adzuki bean seeds. Special attention was given to the content of GABA, the related GAD and DAO activities, and the microbial safety of the produced sprouts.

## 2. Materials and Methods

### 2.1. Seed Materials

Commercially produced adzuki bean (*Vigna angularis*) variety Kaohsiung 9 seeds (100 seeds weight 16.8 g) were purchased from a local market. The seeds with 15% seed moisture content on dry weight basis were stored at 4 °C before they were used for testing.

### 2.2. Soaking (S) Treatment

30 g of adzuki bean seeds were soaked in running tap water for 16 h at 25 °C (Figure 1). The soaked bean seeds were rinsed in double-distilled water, and then used for the sprouting tests.

### 2.3. High-Voltage Electric Field (HVEF) Treatment

The electric field instrument comprised two parts: the high-voltage power supply (20 kV, 50 Hz, You-Shang Technical Corp, Kaohsiung, Taiwan) and a test chamber equipped with electrodes (Figure 2). The seeds were loaded in a polyethylene tray with a cover of the same material to avoid contact with the electrodes. The untreated control and soaked adzuki bean seeds were exposed to an electric field under a field strength of 2 kV/cm for 10 min. The soaking plus high-voltage electric field (SHVEF) treatment was conducted by soaking the seeds for 16 h, and then subjecting them to 2 kV/cm KVEF treatment for 10 min (Figure 1).

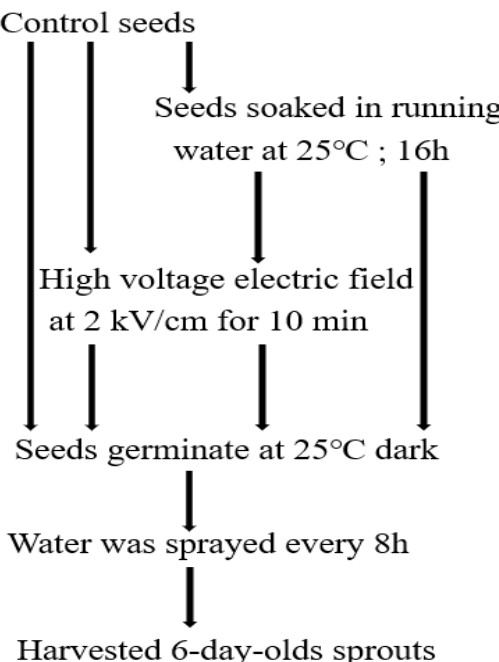

**Figure 1.** Procedure for seed treatments and production of adzuki bean sprouts.

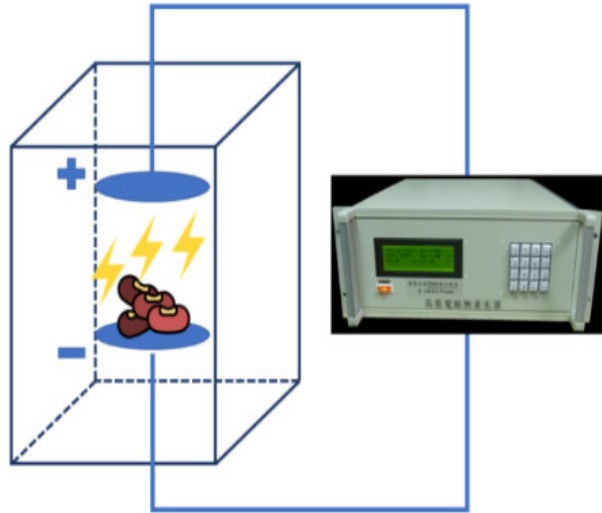

**Figure 2.** Experimental setup for high-voltage electric field treatment. An electric field power supply and a test chamber consisted of two electrodes between which the electric field was generated.

### 2.4. Seed Germination and Hydration Determinations

Five hundred seeds sampled from each treatment were germinated in a growth chamber (ES4-1S, Saint Tien Co., Ltd., Kaoshiung city, Taiwan) in the dark at 25 °C for 6 days. The germination percentage and mean germination time (MGT) were calculated according to the formula of Ellis and Roberts [29]. For water uptake determination, 30 seeds were weighed to obtain the initial weight and then, weights were measured at the following periods: 20, 40, 60, 80, 100, and 120 h. The moisture uptake was determined as a function of seed fresh weight increase and seed weight difference at each sampling time. A subsample of 10 g of the seeds from each treatment was germinated under similar conditions for 6 days, and the growing sprouts were collected for sprout length and sprout fresh weight measurements. A portion of sprouts were washed with double-distilled water, and then freeze-dried. The freeze-dried sprouts were stored for further biochemical analyses.

## 2.5. Scanning Electron Microscopy Examinations

The control and treated adzuki bean seeds were cut with a razor blade, and were attached to stabs using a double-sided adhesive tape. The prepared samples were coated with 30 nm of gold using a Hummer-Technics V sputter coater. The coated samples were scanned and examined using a HITACHI S-3400N microscope operating at 30 kV.

## 2.6. GABA and Glutamic Acid (Glu) Measurements

The freeze-dried sprouts samples were ground to powder, then 100 mg of ground sample was extracted with 1.2 mL of water containing trichloroacetic acid (TCA) solution (5%, *v/v*), followed by being vortexed for 10 s. The sample was kept at room temperature for 1 h and centrifuged at $15,000 \times g$ for 15 min under 4 °C, and the supernatant was used for quantitative analysis of GABA and glutamic acid (Standard purchased from Sigma; St. Louis, MO, USA) by HPLC analysis. The Agilent 1200 high-performance liquid chromatograph was equipped with a Zorbax Eclipse AAA analytical column ($150 \times 4.6$ mm i.d., 5 μm, column temperature was maintained at 40 °C) and a UV/Vis detector (338 nm) (Santa Clara, CA, USA) following the procedures detailed by Li et al. [30].

## 2.7. Glutamate Decarboxylase (GAD) and Diamine Oxidase Activities (DAO) Determination

The freeze-dried sprout powder was used for GAD and DAO activities determination. GAD activity (Standard purchased from Sigma; St. Louis, MO, USA) was determined by the method of Zhang et al. [31]. 1 g of freeze-dried sprout powder was extracted with 10 mL 1/15 M potassium phosphate buffer (pH 5.8) containing 2 mM β-mercaptoethanol, 2 mM ethylene diamine tetraacetic acid, 0.2 mM pyridoxal phosphate. The homogenate was centrifuged at $10,000 \times g$ for 20 min at 4 °C. The supernatant was used as crude enzyme extract. The GAD reaction mixture, which consisted of 200 μL of enzyme extract and 100 μL of substrate (10% Glu, pH 5.8), was incubated in a 40 °C water bath for 2 h, then terminated in a 90 °C water bath for 5 min. The activity of GAD was defined as the release of 1 μmol of GABA produced from glutamate per 30 min at 40 °C.

The DAO activity was determined according to Xing et al. [32]. The crude enzyme extract used for the GAD assay was also used for the DAO assay. Reaction solutions (2.9 mL) contained 2.0 mL of 70 mM sodium phosphate buffer (pH 6.5), 0.5 mL of enzyme extracts, 0.1 mL of horseradish peroxidase (250 U/mL), and 0.2 mL of 4-aminoantipyrine/*N,N*-dimethylaniline. The reaction was initiated by adding 0.1 mL of 50 mM putrescine. The absorbance at 555 nm was read on a UV-2802 spectrophotometer (Unico, Arnold, MO, USA). A 0.01 value of the changes per minute in absorbance at 555 nm was regarded as one unit of the enzyme.

## 2.8. Microbial Analyses

All of the seeds and sprouts samples were analyzed for total aerobic bacteria, total coliform, and total mold [33]. The samples (25 g) were added to 225 mL of 8.5 g $L^{-1}$ NaCl solution, then the solution was homogenized and diluted (a six-fold dilution) with distilled water. The resultant solution was surface-deposited on a Sanita-kun cultural medium sheet (Chisso Corporation, Tokyo, Japan) containing a transparent cover film, an adhesive sheet, a layer of nonwoven fabric, and a water-soluble compound film designed for detection of aerobic microorganisms. The sheets were kept in a temperature-controlled incubator according to the manufacturer's instructions for counting total aerobe bacteria, total coliform, and total mold. Microbial colonies were counted at 35 °C for 48 h, 35 °C for 24 h, and 25 °C for 5 days for total aerobic bacteria, total coliform, and total mold, respectively, after incubation. The results were expressed in $\log_{10}$ CFU $g^{-1}$ fresh weight.

## 2.9. Data Analysis

A complete randomized block design with three replicates was used to evaluate the effects of seed treatments on the tested adzuki bean seeds. The data were presented by using mean ± standard deviation, and statistical significances were determined according

to Duncan's multiple range test. Statistical differences at $p < 0.05$ were marked with consecutive lowercase letters. The correlation coefficients between the examined seed traits were also calculated. The statistical analyses were conducted using IBM SPSS statistics version 19 (SPSS, Inc., Chicago, IL, USA).

## 3. Results

### 3.1. Seed Treatment Effects on Germination and Sprout Growth

Germination is a complex process involving many morphological and metabolic changes in hydrated seeds [34]. Table 1 shows the results of 6-days germination responses observed from the seeds receiving different treatments. Germination percentages were in ascending order of control seeds (82.1%) < HVEF-treated seeds (86.4%) < S-treated seeds (96.3%) < SHVEF-treats seeds (100%). Untreated control seeds showed longer mean germination time (MGT) (3.87 days) compared to the seeds that received different seed treatments (ranged from 1.42 days to 3.43 days) (Table 1). HVEF treatment was reported to increase the hydrophilicity of seeds [35]. The increased seed hydrophilicity would be favorable for seed hydration, and subsequently accelerate seed germination. Moreover, the HVEF treatment may enhance the germination of certain seeds by the activation of various enzymes activities through protein structure modification [20,25]. On the other hand, soaking would soften the seed texture and increase the hydrophilicity, thus, the added effect of soaking and HVEF (SHVEF) treatment would further increase the seed germination to 100% (Table 1).

HVEF is reported to promote the growth of mung bean sprouts [36]. In this study, significant differences in sprout growth existed among the seeds receiving different seed treatments (Table 1). The sprouts subjected to different seed treatments exhibited greater sprout growth and longer sprout length than that of the control group. The sprout fresh weight and length produced from SHVEF-treated adzuki bean seeds were 343.72 g 100 seeds$^{-1}$ and 12.84 cm, respectively, which were 69% and 145% higher than untreated control seeds (203.40 g 100 seeds$^{-1}$ and 5.24 cm) (Table 1), (Figure 3). The exhibited differences in seed germination and sprout growth among control seeds and the seeds receiving different seed treatments are possibly caused by the different water absorption levels observed (Figure 4), which might have initiated metabolic activities that prepare sprouts for growth [19].

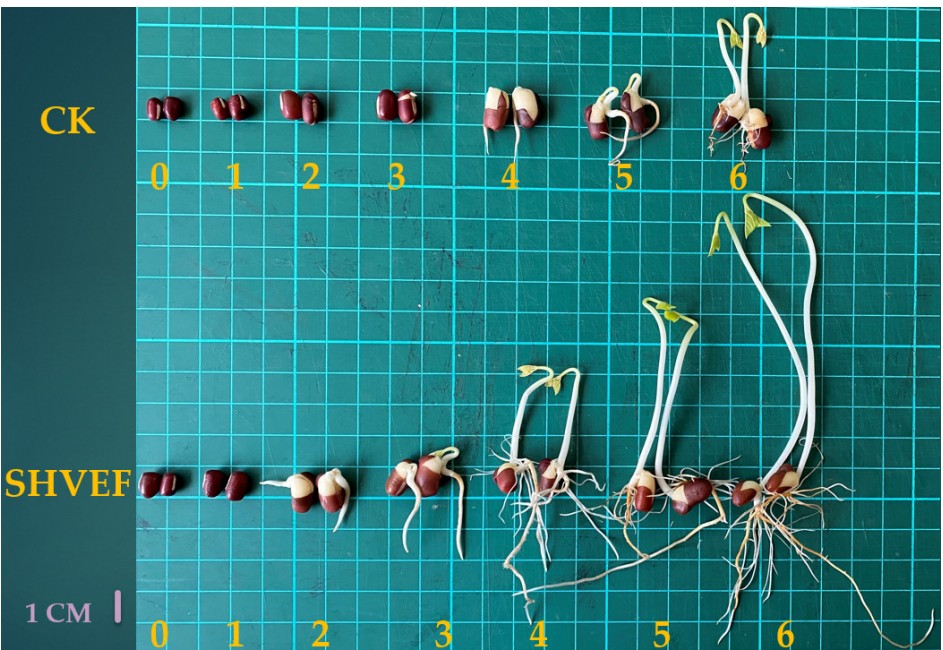

**Figure 3.** Development of untreated control (CK) and SHVEF treated adzuki bean sprouts sampled from day 0 to day 6.

**Table 1.** Effects of different treatments (CK, control; HVEF, high-voltage electric field; S, soaking; SHVEF, soaking + highvoltage electric field) on the germination percentage, mean germination time, sprout fresh weight, and sprout length of 6-day-old adzuki bean sprouts.

| Treatments | Germination | MGT | Sprout Fresh Weight | Sprout Length |
|:---:|:---:|:---:|:---:|:---:|
| | % | days | g 100 seeds$^{-1}$ | cm |
| CK | 82.1 ± 2.12 [d] | 3.87 ± 0.45 [a] | 203.40 ± 2.91 [d] | 5.24 ± 0.32 [c] |
| HVEF | 86.4 ± 0.83 [c] | 3.43 ± 0.17 [b] | 227.82 ± 3.52 [c] | 6.91 ± 0.41 [b] |
| S | 96.3 ± 0.61 [b] | 2.08 ± 0.21 [c] | 284.15 ± 4.71 [b] | 11.27 ± 0.62 [a] |
| SHVEF | 100.0 ± 0.00 [a] | 1.42 ± 0.13 [d] | 343.72 ± 5.83 [a] | 12.84 ± 0.83 [a] |

The data shown are mean values ± SD ($n$ = 3); values followed by the same letter within the column were significantly different ($p < 0.05$).

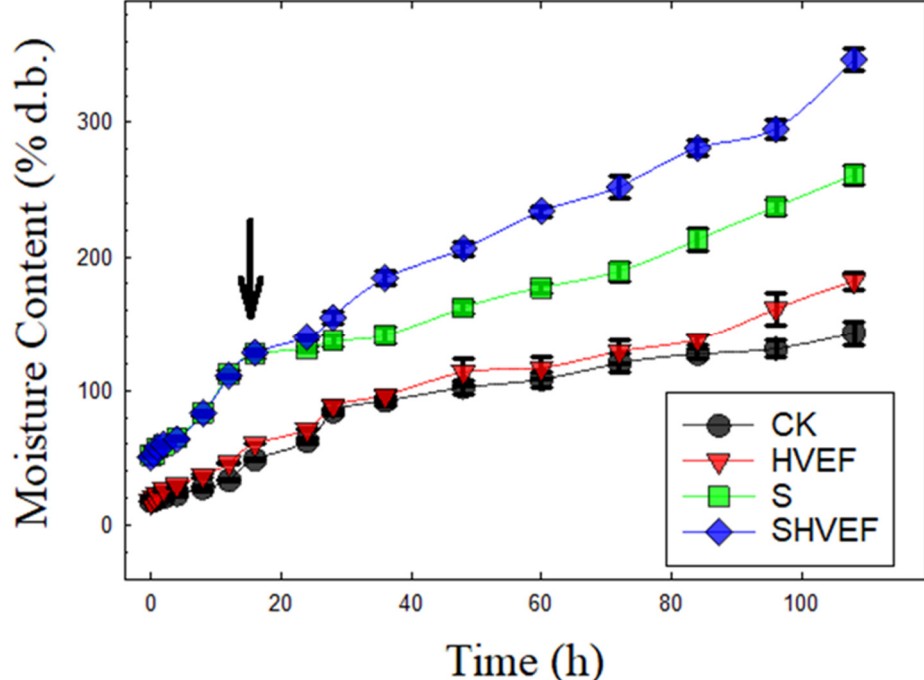

**Figure 4.** Water absorptions of control seed (CK); soaking-treated seed (S); high-voltage electric field-treated seed (HVEF); soaking + high-voltage electric field-treated seed (SHVEF).

### 3.2. Seed Treatment Effects on Seed Hydration

Hydration is the first step toward seed germination. Sufficient water is necessary for reactivating enzymes in preparation for seed germination [19]. During seed hydration, adzuki bean seeds exhibit an initial lag phase, which is characterized by low water absorption, then followed by rapid water absorption [37]. In this study, similar seed hydration patterns were also found in the control seeds (Figure 4). The observed slow initial water uptake was possibly related to the hard seed coat characteristic of the adzuki bean seeds [38]. Once the seed coat is hydrated to some extent (e.g., soaking the seeds in running water for 16 h), its resistance to the water flow may decrease, then the water uptake rate may progressively increase [37].

As shown in Figure 4, significant differences in hydration level existed among the tested adzuki bean seeds receiving different seed treatments, with the control seeds exhibiting significantly lower water uptake than the seeds receiving different seed treatments. Soaking the seed in running water for 16 h appeared to increase seed moisture level and reduce the seed coat's resistance to water flow. Therefore, the seed moisture level of seeds receiving soaking treatment was considerably greater than the seed moisture level of untreated control seeds at the end of 110-h hydration period. Limited studies were available to show the effect of electric fields on seed hydration. HVEF treatment was reported to

accelerate the seed hydration in oat [20] and chickpea seeds [39]. In this study, the seed hydration levels at the end of 110 h of water uptake were in descending order of SHVEF-treated seeds (310%) > S-treated seeds (220%) > HVEF-treated seeds (180%) > untreated control seeds (130%) (Figure 4).

Seed coat is the major barrier for water intake during legume seed hydration. In this study, scanning electron microscopy examinations revealed that the surfaces of SHVEF-treated adzuki bean seeds tended to have more surface cracks (Figure 5D) than control seeds (Figure 5A). These visible cracks appear to be favorable for exchanging oxygen and water between the seed and the external environment. As a result, the seeds subjected to SHVEF treatment absorbed more water than control seeds during hydration (Figure 4). Cold plasma-treated soybean seed was reported as being subjected to an attack by cold plasma-induced oxygen radicals and ions, which resulted in seed coat cracking and erosion during water uptake [40,41]. The altered seed coat surface would increase its hydrophilic ability and ultimately improve the water uptake of the seed. These free radicals and ion attacks on seed coats might also take place on the HVEF-treated seed, because the water absorption of adzuki bean seeds that were subjected to HVEF and SHVEF treatments increased substantially during seed hydration (Figure 4).

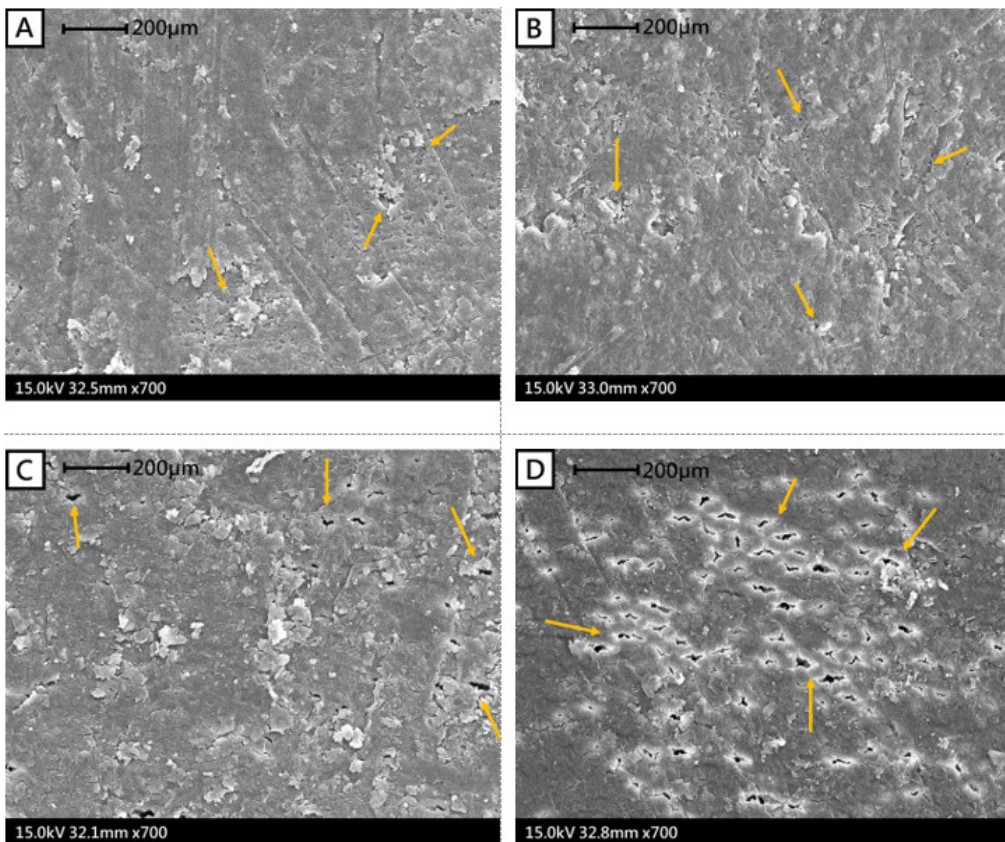

**Figure 5.** The microstructure (SEM, 15 kV, 700 X) of seed coat surface of control seed (**A**); soaking-treated seed (**B**); high-voltage electric field-treated seed (**C**); soaking + high-voltage electric field-treated seed (**D**). The arrows indicate treatment-induced cracks on the seed coat surface.

### 3.3. Seed Treatment Effects on Biochemical Traits

Sprouting is an effective approach to stimulate GABA synthesis [42]. Results of GABA and Glu content and GAD and DAO activities in the tested adzuki bean sprouts are displayed in Table 2. The nongerminated control seeds had extremely low levels of GABA (4.91 mg 100 g$^{-1}$), but the levels increased considerably during sprouting. The level of GABA in untreated 6-day-old sprouts significantly increased to 126.30 mg 100 g$^{-1}$, which is slightly lower than the level (150 mg 100 g$^{-1}$) reported by Li et al. [17]. The Glu levels

also increased during sprouting, even though it was slightly reduced on 4-day-old sprouts (Table 2). But the magnitude of the increase in Glu level was significantly lower than that of GABA (Table 2).

**Table 2.** Effects of soaking (S), high-voltage electric field (HVEF), and S+HVEF(SHVEF) treatments on the changes in γ-aminobutyric acid (GABA) and glutamic acid (Glu) contents, and glutamate decarboxylase (GAD) and diamine oxidase (DAO) activities of adzuki bean sprouts during 6-day sprouting.

| | | Days | | | |
|---|---|---|---|---|---|
| | **Treatments** | **0** | **2** | **4** | **6** |
| GABA (mg 100 g$^{-1}$) | CK | 4.74 ± 0.20 [a] | 14.22 ± 0.31 [d] | 70.31 ± 2.32 [d] | 126.30 ± 2.62 [d] |
| | HVEF | 4.92 ± 0.26 [a] | 24.91 ± 0.92 [c] | 94.62 ± 4.21 [c] | 162.12 ± 4.13 [c] |
| | S | 5.02 ± 0.17 [a] | 42.31 ± 0.61 [b] | 121.11 ± 3.13 [b] | 187.21 ± 3.82 [b] |
| | SHVEF | 5.49 ± 0.24 [a] | 94.80 ± 0.82 [a] | 235.53 ± 4.82 [a] | 258.13 ± 5.64 [a] |
| Glu (mg 100 g$^{-1}$) | CK | 58.72 ± 0.41 [a] | 233.48 ± 7.62 [c] | 208.15 ± 6.81 [c] | 296.46 ± 8.42 [c] |
| | HVEF | 60.23 ± 0.52 [a] | 354.21 ± 8.91 [c] | 316.38 ± 9.42 [c] | 328.76 ± 6.81 [b] |
| | S | 56.85 ± 0.32 [a] | 445.63 ± 9.23 [b] | 357.11 ± 8.31 [ab] | 345.24 ± 7.42 [ab] |
| | SHVEF | 58.24 ± 0.23 [a] | 488.65 ± 6.71 [a] | 387.37 ± 7.92 [a] | 368.46 ± 6.22 [a] |
| GAD (U mg$^{-1}$) | CK | 0.21 ± 0.00 [a] | 1.52 ± 0.00 [d] | 3.41 ± 0.01 [c] | 4.52 ± 0.01 [c] |
| | HVEF | 0.21 ± 0.00 [a] | 2.21 ± 0.01 [c] | 4.72 ± 0.02 [b] | 6.61 ± 0.02 [b] |
| | S | 0.22 ± 0.01 [a] | 2.82 ± 0.01 [b] | 4.51 ± 0.01 [b] | 6.22 ± 0.02 [b] |
| | SHVEF | 0.23 ± 0.01 a | 6.04 ± 0.01 [a] | 7.04 ± 0.04 [a] | 8.13 ± 0.04 [a] |
| DAO (U g$^{-1}$) | CK | 1.12 ± 0.00 [a] | 2.10 ± 0.11 [d] | 4.02 ± 0.11 [c] | 5.21 ± 0.21 [d] |
| | HVEF | 1.12 ± 0.00 [a] | 3.11 ± 0.12 [c] | 4.73 ± 0.20 [c] | 8.52 ± 0.30 [c] |
| | S | 1.13 ± 0.01 [a] | 6.82 ± 0.21 [b] | 10.12 ± 0.31 [b] | 11.42 ± 0.21 [b] |
| | SHVEF | 1.14 ± 0.00 [a] | 9.41 ± 0.13 [a] | 13.92 ± 0.42 [a] | 14.71 ± 0.32 [a] |

The data shown are mean values ± SD (*n* = 3); values followed by the same letter within the column were significantly different (*p* < 0.05).

GABA synthesis requires glutamic acid decarboxylase (GAD) by using Glu as substrate [1]. The activity of GAD was reported to increase significantly during seed germination [42]. Similar results were also observed in this study (Table 2). The GAD activity was increased from 0.21 U mg$^{-1}$ to 4.50 U mg$^{-1}$ at the end of 6 days of sprouting. Research evidence has shown that GABA can also be formed from polyamine degradation, where DAO is the key enzyme [3]. In this study, the DAO activity also increased, but with less magnitude (from 1.12 U g$^{-1}$ to 5.21 U g$^{-1}$ compared to the increase in GAD activity, suggesting that DAO might not be the major enzyme for GABA synthesis during 6-day sprouting (Table 2). As was shown in Table 2, all the seed treatments affected the levels of GABA and Glu, as well as GAD and DAO activities in the produced adzuki bean sprouts. The highest levels of biochemical traits were found in the SHVEF-treated sprouts, followed by S-treated sprouts (Table 2). These results further confirm that the SHVEF treatment should be considered for adzuki bean seed pregermination treatment.

Correlation analyses between sprout fresh weight, GABA and Glu contents, and GAD and DAO activities showed positive relations in the tested adzuki bean sprouts receiving different seed treatments (Table 3), with correlation coefficient (r) values ranging from 0.3727 (significant at 5% level) (GABA vs. Glu) to 0.9247 (significant at 1% level) (GABA vs. sprout fresh weight). The results suggest that the treated (S or HVEF) adzuki bean seeds whose sprouts grew better during soaking are promising for higher GABA accumulation. GABA content and GAD activity were positively correlated (r = 0.8918, significant at 1%) (Table 3). However, their coefficient of determination value (r$^2$) was only 0.7953, suggesting that GABA content in adzuki bean sprout tissue was mainly affected by GAD activity (79.53%), but other factors such as Glu content and DAO (account for 20.47% of determination) activity might also affect the level of GABA to some extent.

**Table 3.** Correlation coefficients between γ-aminobutyric acid (GABA) and glutamic acid (Glu) contents, glutamate decarboxylase (GAD), diamine oxidase (DAO) activities, and sprout fresh weight in the tested adzuki bean seeds subjected to different seed treatments.

| | Correlation Coefficient (r) | | | | |
|---|---|---|---|---|---|
| | GABA | Glu | GAD | DAO | Sprout weight |
| Glu | 0.3727 [*] | - | - | - | - |
| GAD | 0.8918 [**] | 0.6184 [**] | - | - | - |
| DAO | 0.8777 [**] | 0.5713 [**] | 0.8613 [**] | - | - |
| Sprout weight | 0.9247 [**] | 0.5216 [**] | 0.9114 [**] | 0.8484 [**] | - |

[*],[**] Significant at $p < 0.05$ and $p < 0.01$, respectively.

*3.4. Seed Treatment Effects on Microbial Loads in Seeds and Sprouts*

There has been an increasing trend in the consumption of sprouts worldwide; therefore, sprout contamination has become a major concern [26]. The results of the microbiological analyses of tested seeds and produced sprouts are displayed in Table 4. The dry adzuki bean seeds exhibited total aerobic bacteria counts, total coliform counts, and total mold counts of 1.94, 1.08, and 1.78 $\log_{10}$ CFU $g^{-1}$ fresh weights, respectively (Table 4). These values were in the ranges of 1.82, 1.61, and 1.68 $\log_{10}$ CFU $g^{-1}$ fresh weight for total aerobic bacteria, total coliform, and total mold counts, respectively, as reported in adzuki bean seeds previously [43]. As expected, sprouting significantly increased the microbial loads of bean sprouts produced compared to untreated control seeds (Table 4). The increased microbial loads in the produced adzuki bean sprouts are mainly attributable to the microbes present on the seeds and the favorable environmental conditions (e.g., soaking water quality and sprouting temperature) in which they are grown [32].

**Table 4.** Effects of soaking (S), high-voltage electric field (HVEF), and soaking plus high-voltage electyric field (SHVEF) treatments on the microbial loads of adzuki bean seeds and 6-day-old adzuki bean sprouts.

| Treatments | Total Aerobic Bacteria Count | | Total Coliform Count | | Total Mold Count | |
|---|---|---|---|---|---|---|
| | (log $_{10}$ CFU $g^{-1}$ Fresh Weight) | | | | | |
| | Seeds | Sprouts | Seeds | Sprouts | Seeds | Sprouts |
| CK | 1.94 ± 0.13 [a] | 9.39 ± 0.33 [a] | 1.08 ± 0.05 [a] | 6.24 ± 0.11 [a] | 1.78 ± 0.13 [a] | 8.94 ± 0.31 [a] |
| HVEF | 0.95 ± 0.01 [c] | 2.83 ± 0.21 [c] | 0.48 ± 0.03 [c] | 1.32 ± 0.02 [c] | 0.71 ± 0.03 [c] | 2.74 ± 0.12 [c] |
| S | 1.23 ± 0.13 [b] | 4.91 ± 0.41 [b] | 0.78 ± 0.11 [b] | 2.33 ± 0.04 [b] | 1.04 ± 0.02 [b] | 3.47 ± 0.21 [b] |
| SHVEF | 0.31 ± 0.00 [d] | 1.85 ± 0.13 [d] | 0.00 ± 0.00 [d] | 0.00 ± 0.00 [d] | 0.30 ± 0.03 [d] | 1.48 ± 0.13 [d] |

The data shown are mean values ± SD ($n = 3$); values followed by the same letter within the columm were significantly different ($p < 0.05$).

Many physical, chemical, and biological methods can decrease the microbial load of bean seeds [26,44]. In this study, the decontamination efficacies of S, HVEF, and SHVEF treatments on treated seeds as well as the subsequently produced sprouts were compared (Table 4). The decontamination efficacy on treated seeds and produced sprouts was significantly different among the tested treatments. Soaking treatment has been reported to show little effect on reducing microbial population, even though it improves seed germination [27,45], unless the seeds are soaked in running water [46,47]. Our data confirm their findings (Table 4). High-voltage electric field treatment is also an effective approach for microbial inactivation in germinating seeds [48]. The application of electrical fields causes a buildup of electrical charges at the membrane, and subsequently causes the disruption of cell membranes of the microorganisms attached to germinating seeds and sprouts [48]. As a result, the microbial loads in HVEF-treated sprouts were significantly reduced (a 5-log reduction) [49] compared to that of controls (Table 4). It appears that decontaminating the seeds by using a single treatment, such as HVEF, could significantly reduce microbial populations, but SHVEF is more effective (Table 4).

## 4. Conclusions

This study provides information on morphological, biochemical, and microbial statuses of adzuki bean seeds subjected to S, HVEF, and SHVEF treatments. The results indicated that the SHVEF-treated adzuki bean seeds exhibited the greatest sprout growth with the highest sprout GABA content. SHVEF treatment also achieved a 5-log reduction in the microbial loads of total aerobic bacteria counts, total coliform counts, and total mold counts on the produced adzuki bean sprouts. Therefore, SHVEF treatment can be used to increase adzuki bean sprout growth, and provide an effective intervention technique against microbial contamination for adzuki bean sprout production.

**Funding:** This research received no external funding.

**Institutional Review Board Statement:** Not applicable.

**Informed Consent Statement:** Not applicable.

**Data Availability Statement:** The data presented in this study are available on request from the corresponding author.

**Conflicts of Interest:** The author declares no conflict of interest.

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
