# Peer review of "The Changes in GABA, GAD and DAO Activities, and Microbial Safety of Soaking- and High Voltage Electric Field-Treated Adzuki Bean Sprouts"

_agriculture, doi:10.3390/agriculture12040469_

Round 1

Reviewer 1 Report

Abstract

OK

Introduction

OK

Method

OK

Line 134 – “one” change to “1”

Line 177 – SHVEF-treated seeds

Line 180 – HVEF only produced 86.4% sprouting success, but discussion is directed towards this treatment. I think it is better to discuss why SHVEF can achieve 100% sprouting rate too.

Line 193 – This sentence needs a reference. A little bit predictive as well as we can’t be sure water absorption level is the factor responsible for all the changes. EDIT: Ok, you did moisture content study in Figure 4. Maybe you can update by sentence to reflect this?

Line 198 – Formatting – better to have the significance letters as superscript

Line 202 – Figure 3 – what means by those square boxes? Is it equivalent to 1 cm each? Labelling would make the Figure better

Line 218 – would be better to calculate the slope of the graph to show the lag and exponential phase of CK. As time after 50 hours showed reduced in absorption

Line 220 – again, a rate/slope would show absorption is steadily increasing because it seems plateauing to me after 50 hours

Line 234 – the results seem to contradict the statement at line 180. Maybe the author can discuss why

Line 243 – did cold plasma treatment increase the water uptake as well as a result of the cracks?

Line 292 – remove “not very high value but”

Line 299 – remove “which may”

Author Response

My answer are

Point 1: Line 134 – “one” change to “1”

Response 1: I have already corrected it.

Point 2: Line 177 – SHVEF-treated seeds

Response 2: I have already corrected it.

Point 3: Line 180 – HVEF only produced 86.4% sprouting success, but discussion is directed towards this treatment. I think it is better to discuss why SHVEF can achieve 100% sprouting rate too.

Response 3: Soaking-treated seeds can make adzuki bean seed coat soften and hydrophilic, and it is beneficial to HVEF-treated seed coat perforation.

Point 4: This sentence needs a reference. A little bit predictive as well as we can’t be sure water absorption level is the factor responsible for all the changes. EDIT: Ok, you did moisture content study in Figure 4. Maybe you can update by sentence to reflect this?

Response 4: We can refer to “Siddique, A. and Kumar, P. 2018. Phyiolgical and Biochemicals basis of Pre-Sowing soaking seed treatment- An Overview. Plant Archives Vol. 18 No. 2, 2018 pp. 1933-1937.”  [19]

When a dry seed is soaked in water, the uptake ofwater start in tirphasic stages (Bewley, 1997) in which imbibition is the first stage where due to low water potential the rapid water uptake starts. Mc Donald, (2000) reported, DNA and mitochondria are repaired and proteins are synthesized using exist (mRNA). Study regarding the synthesis of RNA in primed seed confirmed by reverse transcription Polymerase chain reaction (PCR) analysis in cotton (Shinde, 2008). While in second stage, water uptake decreased in comparison to the first stage in seed. During this phase, embryo prepared for germination along with the synthesis of mitochondria and proteins by mRNA. The third stage of imbibition process is identified with a rapid increase of water and completed with the rupturing of seed coat and emergence of radical.

Stage first and second is the base phase of seed priming treatment where seed is just few steps away from radical protrusion. It has been found that pre-sowing soaking /hardening of seeds with distilled water, increased germination percentage and seedling vigor in wheat and rice crop (Al Ansari, 1997; Jaiswal et al., 1997 and Suksoon et al., 1998). Job et al. (2000) reported that enzymes involved in the mobilization of storage proteins are either synthesized or activated during seed priming treatment. Enzymes which are involved in the mobilization of carbohydrate (a-Amylase and b-Amylase) and lipid mobilization are also activated during seed priming (Sung and Chang, 1993). 

Point 5: Line 198 – Formatting – better to have the significance letters as superscript

Response 5: I have already corrected it.

Point 6: Line 202 – Figure 3 – what means by those square boxes? Is it equivalent to 1 cm each? Labelling would make the Figure better

Response 6: I have already corrected it.

Point 7: Line 218 – would be better to calculate the slope of the graph to show the lag and exponential phase of CK. As time after 50 hours showed reduced in absorption 

Response 7: Can’t not calculate the slope because it will used the polynomial regression analysis and re-draw the figure.  But I appreciate the reviewer’e comments in this regard.  

Point 8: Line 220 – again, a rate/slope would show absorption is steadily increasing because it seems plateauing to me after 50 hours

Response 8: Can’t not calculate the slope because it will used the polynomial regression analysis and re-draw the figure.  But I appreciate the reviewer’e comments in this regard.

Point 9: Line 234 – the results seem to contradict the statement at line 180. Maybe the author can discuss why  

Response 9: deleted the sentence

Point 10: Line 243 – did cold plasma treatment increase the water uptake as well as a result of the cracks? 

Response 10: Yes

Holc, M.; Gselman, P.; Primc, G.; Vesel, A.; Mozetiˇc, M.; Recek, N. Wettability and Water Uptake Improvement in Plasma-Treated Alfalfa Seeds. Agriculture 2022, 12, 96. [https:// doi.org/10.3390/agriculture12010096]

Point 11: Line 292 – remove “not very high value but”=

Response 11: Yes

Point 12: Line 292 – remove “which may”=

Response 12: Yes

Reviewer 2 Report

The author has been revised the manuscript properly.

Author Response

Thanks for your corrections and recommendations.

I have already corrected it.

Reviewer 3 Report

The topic considered is an interesting one. It showed us SHVEF treatment was effective for increasing the adzuki bean sprout production, improve the nutritional quality and provide an intervention technique against microbial contamination on produced sprouts compared to soaking (S), high voltage electric field (HVEF). But I think that the paper needed to be minor revised

1.The related information about GABA, sprout, soaking and high-voltage electric field (HVEF) were introduced in introduction, but there is nothing about microbial safety in the produced sprouts why you focused on.

  1. Figure 1 and 3 is not clear.

3.There are mistakes in graphical references in the text 3.1 and 3.2. It should be Figure 3A-E about seed hydration but not Fig. A-E.

  1. There is no data about microbial loads in table 3, so how to get the result of 3.4 and analyze seed treatment effects on microbial loads in seeds and sprouts?
  2. It will be better if you can provide pictures of the bean sprouting process corresponding to the data in the table.

Author Response

Thanks for your corrections and recommendations.

I have already corrected it.

Point 1: The related information about GABA, sprout, soaking and high-voltage electric field (HVEF) were introduced in introduction, but there is nothing about microbial safety in the produced sprouts why you focused on.

Response 1: I have already corrected it.

Point 2: Figure 1 and 3 is not clear.

Response 2: I have already corrected it.

Point 3: There are mistakes in graphical references in the text 3.1 and 3.2. It should be Figure 3A-E about seed hydration but not Fig. A-E.

Response 3: I have already corrected it.

Point 4: There is no data about microbial loads in table 3, so how to get the result of 3.4 and analyze seed treatment effects on microbial loads in seeds and sprouts?

Response 4: I have added it.

Point 5: It will be better if you can provide pictures of the bean sprouting process corresponding to the data in the table.

Response 5: I have added it.

This manuscript is a resubmission of an earlier submission. The following is a list of the peer review reports and author responses from that submission.

Round 1

Reviewer 1 Report

Abstract – well written

Introduction – well written with clear direction. However, it is recommended to include more information on GAD and DAO pathways, as only GABA is being explained in detail.

Line 75 – you mentioned DAO activities are also in the focus. But the title of the paper did not reflect that

Based on my limited reading, DAO is mainly present in humans/animals but not plants. Maybe the author can explain more on the mechanism of plant GABA production

Materials and methods

Line 82 – thirty should be a number (30)

Line 88 – is “w” equivalent to “with”?

Line 144 – need superscript

Results

Line 249 – r value of GABA vs Glu (0.3727) is quite low, so the sentence might need to reflect that. Since the precursor of GABA is glutamate, this is quite surprising. It shows that sprout fresh weight is more important for GABA synthesis. Maybe the author can explain why. I also noticed the next sentence using r2 value (0.7953), which is still quite low. R value was shown in Table 3, but not r2 so no meaningful conclusion can be made

Line 259 – I think it should be Table 4, which is missing from the manuscript

Line 265, 272, 276 – Is Table 1 showing any microbial load or effect of decontamination?

I think the author refers to the wrong Table in the microbial study

Figures and Tables

Line 292 - Figure 3 should be split into 2 figures as it shows different outcomes/procedures. A higher resolution is desirable. The SEM figure should be labelled so readers understand where to look for it. At what stage of growth these figures were taken?

Line 304 – GABA initial contents were all 4.91 mg 100g (although the SD is different). Is it possible? Since HPLC can be very sensitive. The unit for DAO is U g, not U mg?

Line 313 – the abbreviation was S+HVEF, but here it is written as SHVEF. Need to be consistent

Author Response

Thanks for your corrections and recommendations.

Reviewer 2 Report

Line 162, Results and discussion is correct. 

The polyamines especially putrescine content should be analyzed for better understand of mechanism.

Author Response

Thanks for your corrections and suggestions

Reviewer 3 Report

In this manuscript, the author reported the effect of high voltage electric field treatment on germination, GABA metabolism and microorganism containing. The story is interesting and the data are helpful for optimizing sprout producing technique. Nevertheless, the writing quality of the manuscript is far less satisfying. Many mistakes exist in the manuscript, as listed below. The author should check and revise the manuscript carefully.

  1. In Part 3.1 and 3.2, the author mentioned Fig.1 A,1 B……,however,Fig.1 in page 7 described experimental protocol only. It looks that all Fig.1 should be Fig.3.
  2. In Part 3.4, the author mentioned Table 1 or Table 3, however, the data in Table 1 and Table 3 did not show the result of microbial load in sprouts. It seems that there should be a Table 4, the data about microbial load was missed.
  3. The resolution of images in Fig.1, Fig.2, Fig.3 are too low and highly blurred.
  4. The author should move the figures to the place near the corresponding text rather than put them together at the end of the manuscript.
  5. Line 130, “ml” should be corrected as “mL”. Line 144, L-1, “-1” should be superscript

Author Response

Thanks for your corrections and suggestions。

Reviewer 4 Report

The topic considered is an interesting one. It showed us SHVEF treatment was effective for increasing the adzuki bean sprout production, improve the nutritional quality and provide an intervention technique against microbial contamination on produced sprouts compared to soaking (S), high voltage electric field (HVEF). But I think that the paper needed to be minor revised.

1.The related information about GABA, sprout, soaking and high-voltage electric field (HVEF) were introduced in introduction, but there is nothing about microbial safety in the produced sprouts why you focused on.

2.Figure 1 and 3 is not clear.

3.There are mistakes in graphical references in the text 3.1 and 3.2. It should be Figure 3A-E about seed hydration but not Fig. A-E.

4.There is no data about microbial loads in table 3, so how to get the result of 3.4 and analyze seed treatment effects on microbial loads in seeds and sprouts?

5.It will be better if you can provide pictures of the bean sprouting process corresponding to the data in the table.

Author Response

(The authors gave the same response as above.)
